# The Role of Culture in Maintaining Post-Partum Sexual Abstinence of Swazi Women

**DOI:** 10.3390/ijerph16142590

**Published:** 2019-07-20

**Authors:** Zinhle Shabangu, Sphiwe Madiba

**Affiliations:** Department of Public Health, Sefako Makgatho Health Sciences University, Pretoria 0001, South Africa

**Keywords:** Eswatini, cultural beliefs, sexual abstinence, post-partum, social norms, practices, Theory of Planed Behaviour

## Abstract

Eswatini is one of the countries in the African continent where post-partum sexual abstinence is practiced. Beside scarcity of research exploring sexual abstinence in Eswatini, there are only a few studies that explore post-partum abstinence across HIV-positive and negative women in sub-Saharan Africa. The study explored the practice of post-partum sexual abstinence in Swazi women and examined how cultural beliefs influence and promotes the perpetuation of the practice. The study population consisted of post-partum women who were selected, using purposive sampling. Thematic approach was used for data analysis. Despite feeling that the period for post-partum, sexual abstinence was long; the participants adhered to the practice as prescribed by their culture. Nevertheless, they felt that the practice is imposed on women only because while they are observing post-partum abstinence, their partners get to sleep with other sexual partners. They raised concerns that the practice increases the risk of acquiring HIV and sexually transmitted infections. There is an element of coercion to the practice of post-partum abstinence, the myths and misconceptions around the early resumption of sexual intercourse forces the practice on women. At the family and community level, the discussions to change the way sexual abstinence is viewed and practiced are crucial.

## 1. Background

The International Commission of Fertility (IFC, 2014), define post-partum sexual abstinence as a period of voluntary sexual abstinence after pregnancy and delivery. Yet in many societies in the African continent and other regions, the practice of prolonged post-partum sexual abstinence is a strong cultural practice [1,2,3,4]. Literature shows that the period for abstinence and resumption of sexual activities is influenced by a variety of factors and varies from culture to culture [5,6]. Research indicates that often the resumption is prescribed by the needs of the baby, the father, the age of the mother, the marital status, and cultural beliefs [1,5,6]. For instance, the younger the mother, the earlier the resumption of sexual intercourse because young mothers claim that they do not know any rules on sexual abstinence during the post-partum period [7,8,9]. With regards to marital status, married women tend to stick to the culture of abstinence after delivery, as compared with single women [8].

In many settings in the African region the duration of post-partum abstinence ranges from three months to over one year and sometimes the period is almost equivalent to when the child is able to walk or until the child is weaned from the breast [5,6,8,10,11,12]. However, a report by the World Health Organization indicates that the period of post-partum abstinence is on the decrease globally [13]. For example, in recent studies conducted in Tanzania and Uganda, sexual abstinence for most women ends at five to six weeks post-partum [5,14]. In a study conducted in Kenya the median time of resumption sexual activity after delivery was seven weeks [15].

Many cultural related factors attribute to the decrease in the period of post-partum abstinence. In the African context, polygamous marriages created an environment where the men had other wives with whom to have sexual intercourse when one of the wives was abstaining [3,16]. However, marriages are monogamous now, and the prevalence of polygyny has declined over the last century in many African nations [17]. For example, data from 34 African countries show that 72% of women were in monogamous marriages, 19% report that their husband has two wives, 7% report that he has three wives, and fewer than 2% report that he has four wives or more [18]. In monogamous marriages, the couple reside in the same house even when the woman has just delivered, which results in the early resumption of sexual intercourse, as there are no in-laws to enforce abstinence by separating couples [19]. Furthermore, in African societies breast-feeding was compulsory and it was one of the reasons women had to abstain, but nowadays breast feeding is optional, as babies are sometimes fed formula [1,16]. Moreover, modern methods of family planning are used when traditionally, sexual abstinence was a way of avoiding early pregnancy [16,20,21].

Of significance is that researchers argue that the decrease of the period of abstinence could also be influenced by the view that the practice facilitates the spread of human immunodeficiency virus (HIV) [5,22,23]. There is evidence that prolonged post-partum abstinence in the era of HIV increase the risk of HIV acquisition during the post-partum period, relative to the non-pregnant period [24,25]. For example, a meta-analysis found that the risk of mother-to-child HIV transmission is higher during the post-partum period among those who had recently acquired HIV than among those with chronic HIV infection [25]. The literature show that a substantial number of men engage in extramarital sex when their wives practice post-partum abstinence as a way to satisfy their unmet sexual needs. Studies conducted in Nigeria reported that 28%–43.7% of men reported concurrent sex partners during their wives’ pregnancy and post-partum [1,15]. Concurrent sexual partnerships put men, and subsequently their female partners, at risk of acquiring sexually transmitted infections (STIs) including HIV with adverse maternal and fetal outcomes [13,16,20]. 

Consequently, there is a move to promote the early resumption of sexual activities during the post-partum period [3]. The early resumption of sexual intercourse in the post-partum period makes men stay at home more and not engage in sexual intercourse with women other than the wife, reducing the risk of HIV transmission and other STIs [13,16]. On the other hand, early resumption of sexual intercourse may pose health challenges to the mother, as post-partum abstinence is one of the factors determining maternal and child mortality [1,13]. One of the consequences of early resumption includes unwanted pregnancy, which may be undesirable, especially if no contraceptive method is used [16].

Eswatini is one of the countries in the continent where post-partum sexual abstinence is still practiced. The Swazi culture restricts post-partum women from engaging in sexual intercourse with their partners for a period of six months [26]. Culturally, the traditional Swazi family controls, determines, and safeguards all the actions of its members. The decision to resume sex during the post-partum period is also one of the practices that is determined by the family. Nevertheless, there is no formal documentation that to duration of post-partum abstinence in Swazi health literature, even though after delivery, according to Swazi culture, a woman is expected to abstain from sexual intercourse for about six months. 

It is a public health concern that in this era of HIV not much is known about the practice of post-partum sexual abstinence and the issues associated with the resumption of sex among women, when some of them or their partners might be HIV-positive [5,22]. Eswatini has the highest HIV prevalence in the world, the Joint United Nations Programme on HIV/AIDS estimates that 27.4% of adults were living with HIV in 2017 and women are disproportionately affected by the HIV epidemic [27]. As already said, there is evidence that the risk of HIV acquisition increases throughout pregnancy and is highest during the post-partum period [24]. In the meta analysis referred to previously, the highest HIV incidence estimates during pregnancy/post-partum were from Eswatini and neighboring South Africa [25]. 

The aim of the study was to explore the practice of post-partum sexual abstinence in Swazi women and examine how cultural beliefs influence and promotes the perpetuation of the practice. It is important to understand the sexual behaviors during the post-partum period to develop HIV preventive strategies to mitigate the high HIV incidence observed among women during these periods [15,24]. The study will begin to close the gap in the literature and will inform the sexual health promotion messages and counselling provided to women in the prevention of mother to child transmission of HIV (PMTCT) program.

## 2. Material and Methods

### 2.1. Study Design 

This paper is extracted from the first author’s dissertation, which was submitted in partial fulfilment of the requirements of a Master’s degree in Public Health. The aim of the study was to explore the practices of Swazi women concerning socio-cultural abstinence during the post-partum period.

This paper used the Theory of Planned Behavior (TPB) to explain the adoption of post-partum abstinence practice (Figure 1). The TPB has proven to be a useful model in explaining intentions to engage and actual engagement in a number of health behaviors [28]. The usefulness of the TPB in explaining actual engagement in health behaviors influenced the use of the theory framework in the current study include women who were practicing post-partum abstinence. This is because the study was conceptualized to include women who were practicing post-partum abstinence. Nevertheless, the theory assessed their intentions to maintain abstinence for the expected period. The belief-base framework of the TPB was used to examine the attitudes, the normative, and control beliefs underpinning women’ decisions to practice post-partum abstinence. The TPB posits that intentions to perform behavior can be predicted from attitudes towards the behavior, subjective norms (beliefs about the views of others), and perceived behavioral control (beliefs about factors that facilitate or impede a behavior) [29]. Subjective norms are significant predictors of an individual’s behavioral beliefs and consists of the social pressure to perform or not perform the target behavior [29], in this study, to practice and maintain post-partum sexual abstinence. In addition, the TPB adds perceived control over the behavior, stating that there are situations where an individual may not have complete control over behavior. In the current study, the perceived control on the practice of post-partum sexual abstinence lies outside the control of the women because of its roots in cultural norms and beliefs. 

### 2.2. Study Setting and Population

The study population consisted of post-partum women who were selected using purposive sampling. The study setting was a health facility located in an urban area in the Manzini region, the centre of Eswatini. Eswatini is a land locked country located in the Sub Saharan Region; occupying an area of 17,634 km^2^ with a population of 1,018,449 people [26]. It is bordered by two neighboring countries, South Africa and Mozambique. It is a patriarchal society, which subscribes to strong cultural practices such as polygamy, “Kwendzisa,” and paying Lobola “bride money.” Whereas the prevalence of polygyny has declined over the last century in many African nations [17], around 12% of women aged 15–49 years are in polygamous marriages in the country [30]. Culturally, the traditional Swazi family controls, determines, and safeguards all actions of its members, which necessitates consultation before any member makes major decisions. The Swazis, as a nation, adhere to and maintain a sense of pride in cultural beliefs and practices that have been in place for many generations. The country, however, has a number of potentially high-risk traditions and current practices that make the population vulnerable to HIV infection. This include the clearly defined gender roles for men and women, the subordination of women to men, and the social disempowerment of women and their inability to make decisions about sexual preferences. The engagement of men in multi-concurrent partnerships are relatively high, for instance, in 2010, 16% of men had more than one sexual partner in the last year compared to 2.7% of women [26,30]. 

### 2.3. Data Collection

The first author and a research assistant (the interviewers) with skills in qualitative research methods conducted the interviews using the local language (SiSwati). The interviewers used an in-depth interview schedule with open-ended questions to interview the participants. The guide was developed after reviewing the literature on post-partum abstinence from studies conducted in the region since there are no studies conducted in Eswatini [1,13,24]. The development of the tool was also informed by informal discussions with women who are custodians of culture in the community as well as health professionals. The interview guide consisted of open-ended questions to assess the participants’ views and attitudes about the post-partum abstinence, the way that abstinence is enforced, and the duration of abstinence, experiences with abstinence, and their concerns about the practice. The interview guide addressed each construct in the TPB (behavior of interest, attitudes, subjective norms, perceived behavioral control, and actual behavioral control). Examples of questions are listed in Table 1. 

The use of the local language and the open-ended questions allowed the participants to express their views about sexual abstinence using perspectives unique to their culture. This was facilitated by the fact that the interviewers are Swazi women who can speak and understand SiSwati and the local cultural beliefs well. Each interview was conducted in a private consultation room to ensure privacy, lasted for about 30 min, and was recorded after permission was sought from the participants.

### 2.4. Ethical Considerations

The Ethics Committee of Sefako Makgatho Health Sciences University provided ethical clearance for this study (SMUREC/110/2017: PG). The Ministry of Health in Eswatini granted permission to conduct the study. The participants signed an informed consent form before the interviews took place, they were informed that participation in the study was voluntary and were assured of confidentiality. 

### 2.5. Data Analysis

The audio interviews were transcribed verbatim and translated into English by the research assistant and the first author. Thematic approach was used for data analysis to identify and report patterns or themes within data [31] using both inductive and deductive approach [32]. The deductive approach was used to identify broad descriptive categories based on the TPB constructs and the interview guide. The priori codes were then used to code interviews systematically and grouped the labelled phrases in categories that could be classified into the constructs of the TPB. 

Since deductive analysis provides a less rich description of the overall data but a more detailed analysis of some aspect of the data such as the TPB constructs [31], the inductive analysis approach was used in order to generate codes directly from the data. One of the tenets of inductive thematic analysis is immersion in the data in order to obtain the sense of the whole data through reading and rereading of transcripts [32]. To achieve this, both authors read a few transcripts repeatedly to identify emerging codes to inform codebook development. The development of a codebook was accomplished through a session of reviews to reach consensus on the definition of themes and finalization of the codebook. All the transcripts were uploaded to NVivo 12 (QSR International, Melbourne, Australia), a qualitative data analysis software where further coding was done. Codes were identified, merged, and sometimes changed as new understanding of data emerged. This was done until the authors agreed on the final themes and subthemes that reflected the constructs of the TPH and emerging themes to describe the practice of post-partum sexual abstinence. 

Methods used to ensure rigor included data and investigator triangulation, transcribing verbatim in the language of the interviews, and peer debriefing. The first author spent time in the field to familiarize with the data and the study participants. To ensure credibility of the findings, the two authors took part in the data analysis [33]. 

## 3. Results 

### 3.1. Description of Study Sample

The study sample comprised of 15 post-partum women with a mean age of 27 years, and their ages ranged from 18 to 40 years. Only five of the women had one child, most (8) had between three and six children. At the time of the interview, over half of the women (8) had not resumed sexual intercourse and seven women had resumed, and most of them (5) had resumed when their babies were six months of age. Only three of them were employed, and most (11) had gone up to secondary school. Table 2 present additional sociodemographic of the participants.

### 3.2. Themes:

Four main themes based on the based on the TPB constructs and the interview guide were identified (Table 3). 

#### 3.2.1. Attitude towards Post-Partum Abstinence

The participants were asked for their opinions about sexual abstinence during the post-partum period. They perceived post-partum sexual abstinence as beneficial to them, their babies, and their partners. They said that abstinence facilitates recovery from birth trauma. Furthermore, they felt that abstinence allows the mother adequate time to take care of the baby instead of focusing on the relationship with the partner. The analysis revealed that subjective norms that influence behavior are also present in the belief that post-partum abstinence is beneficial and in motivating the participants to abstain. The TPB states that personal and social beliefs and values determine personal attitudes and perceived social expectations. The participants’ positive attitudes are not necessarily from their personal experiences but are informed by social beliefs and values held by society at large.
“There is something good about the practice. The good thing is that the mother gets to heal very well and take good care of the baby before she resumes sex, before she goes back to the father.”(P 9, 39 years old)
“As a mother you have ample time to give love to your baby, not rushing to focus your attention on the father.”(P 1, 38 years old)
You get to space your children well and your baby grows up well.”(P 4, 41 years old)
“He [the partner] is able to budget on how to raise his children. He does not have to worry about early pregnancies.”(P 2, 32 years old)

#### 3.2.2. Period of Abstinence

The interviews explored the women’s beliefs about the community norms that determine the period of abstinence. The analysis found that the period of abstinence is determined by society through the family and the in-laws who exert pressure on the woman to practice post-partum abstinence. The social pressure to abstain is applied by the broader society that expects women to abstain and even sets the control measures to ensure compliance. The participants also reported that the period of abstinence is also determined by the gender of the baby. When the baby is a girl, the period is longer than when the baby is a boy because of the value that society places on the boy child. The need and the expectations from the women to produce an heir, results in the relaxation of the period of abstinence.
“It depends, with my in-laws you abstain for six months, not sleeping with the father of the baby. You stay for two months, depending on the sex of the baby, if it is a girl you stay indoors for two months in a sacred hut in the homestead, and you don’t sleep with your partner. Even after joining your husband in the house, you continue abstaining until six months.”(P 3, 23 years old)
“My father-in-law called me and my husband and said that in his homestead couples wait six months before they resume sexual intercourse after the birth of a baby.”(P 4, 41 years old)

#### 3.2.3. Beliefs about the Control of Post-Partum Sexual Abstinence

Regarding normative beliefs about community norms and factors that facilitate and perpetuate post-partum abstinence, the participants highlighted cultural norms, not staying with the partner, and myths and misconceptions about early resumption as factors that facilitate the maintenance of sexual abstinence during the post-partum period. 

##### Cultural Norms

The TPB states that various factors can influence perceived behavior control, and culture was identified as one of the factors that influence the post-partum abstinence among the participants. The data demonstrated that culture prescribes and exerts pressure on women to practice sexual abstinence during the post-partum period. According to the TPB, in order to predict the intention to perform a behavior, one needs to know how much the individual feels the social pressure to do it. The participants indicated that cultural norms and expectations influenced their adoption and sustenance of post-partum abstinence. The subjective norms that influence behavior are also present in the adoption of cultural beliefs and practices of post-partum abstinence. This is evident from the narratives of the participants below;
“Long ago, the elders say that the husband used to sleep in his own hut and the wife would have her own where she would sleep with her children. When the husband wanted to have sex with the wife, he would call her to his hut on that particular day (laughs). They said sleeping together every day as a couple you are bound to have sex”(P 1, 38 years old)
“In fact a woman who has delivered in the Swazi culture stays in the sacred hut and stays there for six…, the six months really in the six months he does not see the baby, the father does not have to see the baby before it is six months old”(P 8, 18 years old)

##### Living Apart from the Partner

Living apart from the spouse or the father of the child for single mothers is used as a tool to exert pressure on women to abstain. Culture prescribes that couples married or single should not stay together or have sexual intercourse after childbirth. Whereas for married women the in-laws enforce sexual abstinence, for single women, their mothers exerts the same pressure on them to practice post-partum abstinence. One of the TPB construct is perceived control over the behavior; the analysis demonstrated that the women have no control over whether or not to be with the spouse or the father of the child after childbirth. The participants indicated that they were not living with their spouses or partners during the post-partum period and they believed that staying together would have resulted in the couple resuming sexual relations early after childbirth.
“It is good that after delivery the woman does not stay with the man because one, when you have just delivered you do not have the strength, you need like stay away from him, you do not visit him, but if you stay together you end up resuming, resuming sex early.”(P 6, 30 years old)
“What might make you not to resume early is not staying with him, if you stay far from him you will delay resuming sexual intercourse.”(P 13, 24 years old)

##### Myths and Misconceptions about Early Resumption

Since subjective norms determine an individual’s behavioral beliefs, the cultural beliefs, myths and misconceptions about early resumption of sexual intercourse prevalent in the study setting, influence the adoption and maintenance of post-partum abstinence among the women. The myths and misconceptions enforced sexual abstinence during the post-partum period and the women complied because they had no control of their behavior. The loss of control was a result of the beliefs held by the community about early resumption of sexual intercourse. The participants were told by older women that resuming sexual intercourse early after childbirth affects the spouses or partners adversely; as there are sicknesses that they develop which might even result in them dying.
It is believed that the man is affected and he dies young because he had sexual intercourse with a woman before six months post-partum. This affects him in some way.”(P 2, 32 years old)
“They say he gets ‘ligola’; coughing…I don’t know if it is a chest infection, something like that. Resuming sexual intercourse while the woman is still having a vaginal discharge, you see when she has just delivered.”(P 4, 41 years old)

The baby’s health, growth, and development is believed to be slowed down by the failure of the parents to abstain.
“The baby is said to regurgitate food after eating when the parents fail to abstain during the post-partum period. I don’t know, the reason is that you have resumed sexual intercourse prematurely after the birth of the baby. She regurgitates due to the early resumption.”(P 2, 32 years old)
“They… in the community they tell us that if you do not abstain the baby’s neck will be weak and the baby cannot hold his neck upright. They tell us that the baby does not grow well, it becomes weak.”(P 7, 20 years old)

The participants also believed that sexual intercourse during the post-partum period contaminates breast milk and makes it unhealthy for the baby.
“The sperm goes to the blood and the baby’s milk is made in the blood. I think something happens to the baby’s milk; it becomes either weak or something. Some babies lose weight, I do not know, but I think something does happen.”(P 13, 24 years old)
“I think that…, you see it is like in the hospital when they say we should use a condom. The baby’s milk, I think it makes it weak.”(P 13, 24 years old)

#### 3.2.4. Beliefs about Negative Consequences of Post-Partum Abstinence

The participants were asked to discuss negative beliefs and feelings about post-partum abstinence. They mentioned negative behavioral beliefs related to post-partum abstinence derived from their perceived disadvantages of the practice. They indicated that sexual abstinence is imposed on women and that it increases the risk of HIV transmission. They stated that these are the most significant disadvantages of the practice of post-partum abstinence for women. 

##### Sexual Abstinence Is an Imposed Cultural Practice

Although the participants adhered to the practice of post-partum abstinence, the data showed that some of the women perceived abstinence during the post-partum period as a cultural practice that is imposed on women. As already said, the participants’ positive attitudes towards post-partum abstinence are informed by social beliefs and values held by society and not necessarily from their personal experiences of the practice. Whereas subjective beliefs and attitudes may influence the individual to choose to perform or not perform a behavior, for cultural practices such as abstinence, the control on the practice of post-partum abstinence lies outside the control of the women. Nevertheless, they indicated that abstinence is not a good cultural practice but a practice that they do not like or feel they need.
“It is not bad if it is six weeks as that is what we are given in the hospital, it is right, but the six months that is enforced by my in-laws I find it bad”(Pat 3, 23 years old)
“I think it is a bad culture that women are forced to abstain; they should decide what they want to do in line with what they agree on. According to me it should not be that it is a cultural practice, it should be an agreement between the man and the woman.”(Pat 9, 39 years old)

As already said, the participants felt that post-partum abstinence is a culture imposed on women only. Their narratives suggest that some of the men do not believe in the culture and in delaying resumption of sexual activities after childbirth.
“Men do not believe in delaying sex after childbirth. Even when you have discussed and decided on when to start, you end up not adhering. It is up to you to refuse and refuse but you end up giving in.”(P 6, 30 years old)
“If I had stayed at my parental home, I would have delayed. It is because when you stay with the man it is easy to resume.”(P 2, 32 years old)
“It is because the men cannot be patient for a long time. They then go to other partners.”(P11, 23 years old)

##### Sexual Abstinence Increases the Risk of HIV Transmission

Besides being an imposed cultural practice on women only, the participants said that post-partum abstinence increases the risk of HIV transmission. The fact is that whilst women abstain, men are not required by culture to do the same and often engage in sexual relations with other partners.
“Traditionally it is good, but in these modern times, it is not good because the man can go to other people, to have sex with them because I cannot, he then goes to enjoy with other women”(Pat 1, 38 years old)
“You find that when you return from your parents after delivery at whatever time…, maybe at six months… Many a times you find that there is a baby, your husband has gotten someone pregnant during the period when you were absent from your home.”(P 9, 39 years old)

## 4. Discussion

We explored how culture facilitated sexual abstinence in the post-partum period in a patriarchal society where family values are highly regarded. We found that the participants adhered to the cultural practice of sexual abstinence during the post-partum period. Most of those who had resumed sexual intercourse did so after the six months that is prescribed by culture. Early resumption for a few participants was attributed to the male partner’s demand for sex. The seven women who had not resumed sexual intercourse intended to do so until six months. 

The data revealed that sexual abstinence is considered an important cultural practice among post-partum women in the study setting. This belief was influenced by subjective norms, social beliefs, and values held by society about post-partum sexual abstinence [29]. The participants felt that sexual abstinence was beneficial to the mother, the baby, and the father of the baby. They believed that the early resumption of sexual intercourse during the post-partum period would delay the body’s recovery and the process of healing the birth injuries. Similar findings were reported elsewhere [8]. 

Furthermore, the participants felt that abstinence during the post-partum period allows the mother adequate time to take care of the baby and bond with her baby. They reported that during the abstinence period they were able to attend to their babies’ needs, including breast-feeding, without interruptions from their partners. These findings are in line with those of other studies [16,34].

The participants highlighted the factors that facilitated sexual abstinence. Culture was identified as a social factor that facilitates the adoption and maintenance of sexual abstinence during the post-partum period. The participants indicated that cultural norms and prescriptions assisted them to maintain abstinence. The belief in cultural norms and societal practices facilitated sexual abstinence during the post-partum period. For example, the place of residence is a significant factor in sexual abstinence in the post-partum period [13]. Abstinence is successful in societies where physical and emotional separation was practiced by couples after childbirth [8]. The participants reported that they were separated from their spouses or partners and stayed with parents or relatives during the post-partum period. This practice is observed by married and single women after childbirth. A study conducted in Ghana, found that physical separation of couples reinforced sexual abstinence [10]. 

Although the participants were practicing sexual abstinence because they believed that it was an important cultural practice, they were also influenced by subjective beliefs about myths and misconceptions about the early resumption of sexual intercourse. The most common misconceptions were about the health of the partner and the baby. These misconceptions were used as control of the practice of post-partum abstinence because women will do everything in their power to protect their babies from ill health. For instance, there were beliefs that the early resumption of sexual intercourse in the post-partum period hinders the growth and development of the baby, who will be attacked by illnesses. The baby is believed to be poisoned by breast milk, which mixes with the sperm during sexual intercourse. In their understanding, breast milk is made in the blood, so the two mix and the milk becomes weak. Likewise, studies conducted in Abidjan and Nigeria reported similar beliefs that the mother’s breast milk harm the baby’s health if she resumes sexual activities early [7,8]. In addition, the baby develops excessive regurgitation of food if the parents resume sexual intercourse early in the post-partum period. Some participants thought that the baby becomes so weak that it cannot hold its head up straight. In Tanzania, it is believed that sexual abstinence during the post-partum period is a way of protecting the baby from a sickness called “kubemenda,” which is also caused by the parents of a baby resuming sexual intercourse in the post-partum period [1,35]. The sickness is similar to that described by the participants in the current study.

Likewise, there are beliefs that early resumption might result in the husband’s contracting infections from the woman because she was still “wet,” as in that she was still having a vaginal discharge. One type of infection mentioned by the participants in this study was an illness called “ligola,” a disease that manifests with the husband coughing severely as though he has tuberculosis. It is believed that the disease is so severe that the man might die. In Malawi, women are considered dirty after childbirth until the return of menstruation [20]. There are also derogatory names given to women who fail to maintain sexual abstinence during the post-partum period. To avoid being called such names, women would abstain from sexual intercourse during the post-partum period [1].

Despite the fact that the participants believed that post-partum abstinence was beneficial, they raised several issues about the practice that they did not agree with. They indicated that while they are observing post-partum abstinence their husbands get to sleep with other sexual partners. They had concerns that sleeping with other women would increase the risk of the partner’s contracting diseases like HIV and subsequently bring it home. Their concerns were echoed by researchers in different settings who highlighted that sexual abstinence creates the risk that men will look for sex somewhere else, putting themselves and their partners at risk of contracting STIs and HIV [8,10,23,35,36,37].

Although the participants were practicing sexual abstinence, some perceived abstinence during the post-partum period as a cultural practice that is imposed on women. According to Mbekanga et al. [1], abstinence during the post-partum period is considered a patriarchal cultural practice that oppresses women. Other researchers argue that cultural practices such as post-partum abstinence are gendered [38]. In the past, men married many wives and would engage in sexual intercourse with the other wives whilst the one who had delivered abstained from sexual intercourse. It was only women who were supposed to totally abstain during the post-partum period [34]. While their husbands abstained from sex with their breast-feeding wives but not with other women [3,16,38]. Current findings show that in order to satisfy their unmet sexual needs, men engage in concurrent sexual relationships when their wives practice post-partum abstinence [1,15]. 

The participants indicated that, generally, men do not believe in post-partum sexual abstinence. They narrated incidences where their partners wanted sex immediately after childbirth or demanded sex before the period of abstinence ended. In some cases, despite women having decided to abstain from sexual intercourse during the post-partum period, their partners’ demand for sex would result in them changing their decision and resuming sexual intercourse early [35,39]. In African settings, men are the ones who usually initiate sexual intercourse, sometimes against the will of their partners, who usually give in because they fear that the men will go elsewhere to have sex with other women [8,11,35]. 

The participants wanted sexual abstinence to be an issue for couples and not for the in-laws or the elderly parents. Furthermore, the current discussions about abstinence during the post-partum period involve women only, who are told to abstain, while the men are left to engage in sexual intercourse with other women. Consistent with literature [11,35,38], they believed that the issue could be resolved if their male partners were included in the talks or discussion about abstinence, including talks about the period of abstinence. Based on the TPB constructs, the women suggest that if the subjective norms and perceived behavioral control of society about abstinence change, they will be able to decide whether or not to abstain.

## 5. Conclusions

The study found that post-partum sexual abstinence is practiced by women in the study setting. Despite feeling that the period for post-partum sexual abstinence was long, the women somehow adhered to the practice as expected by cultural and subjective norms in their communities. Most of those who had resumed sexual activity did so after the six months prescribed by culture. The subjective norms and behavioral control are the components of the TPB that have significant influence on post-partum sexual abstinence.

Despite the belief that post-partum abstinence is beneficial, the participants felt that the practice is imposed on women only while the male partners may continue to be sexually active. They raised concerns that it increases the risk of HIV and STIs transmission, because while they are observing post-partum abstinence, their spouses and partners get to sleep with other sexual partners. Nevertheless, those who had not resumed sexual intercourse intended to abstain until the six months that is expected.

Furthermore, the study found that there is an element of coercion to the practice of post-partum abstinence if women would be given derogatory names if they fail to maintain sexual abstinence during the post-partum period. This suggests that the women, who resumed sexual activities earlier than the six months prescribed by culture, did so because they were pressured by the partners. In a patriarchal society, men are the ones who usually initiate sexual intercourse and the woman obeys the husband even though this might have put her in bad light from the in-laws. The belief that the woman is dirty during the post-partum period, the myths, and misconceptions around the early resumption of sexual intercourse force the practice on women.

The findings of the study provide a basis for commencing discussions around sexual abstinence in the era of HIV. At the family and community level, the discussions to change the way sexual abstinence is viewed and practiced are crucial, particularly in the era of HIV, where the couples are at risk of HIV transmission or reinfections. Several factors should be taken into consideration when researchers discuss post-partum abstinence given the changes in the nature of sexual relationships. It should be noted that marriages have become monogamous in many societies, cohabiting is prevalent, and childbirth is common among single women. Therefore, the issue of polygamous marriages cannot be used as the basis of the practice anymore.

It is important that health care professionals be sensitized about the need to discuss abstinence with post-natal women during the post-partum period, particularly among the population of young mothers who may not be able to negotiate sex after childbirth. The involvement of men in the discussions about sexual abstinence during the post-partum period would be beneficial.

## 6. Limitations

The study sample was small and limited to a health facility, therefore the findings cannot be considered to represent a broad section of the women in Eswatini.

## Figures and Tables

**Figure 1 ijerph-16-02590-f001:**
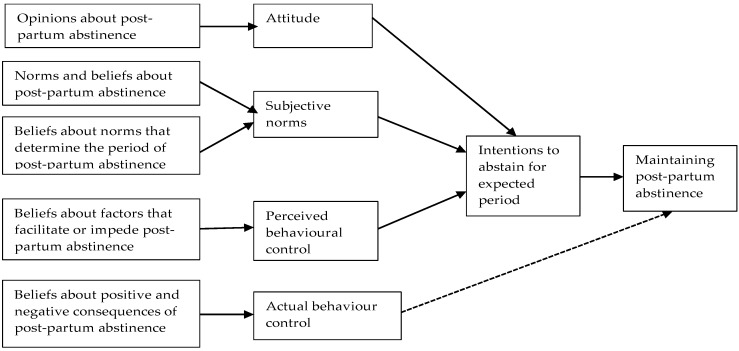
Adapted TPB model on post-partum sexual abstinence.

**Table 1 ijerph-16-02590-t001:** Questions from the interview guide.

TPH Construct	Questions
Opinions about sexual abstinence	What are your views about the practice of post-partum sexual abstinence for women?
Community norms and beliefs about post-partum abstinence	What are some of the discussions around post-partum abstinence that are held in your community?
Beliefs about community norms that determine the period of post-partum abstinence	What or who determines the duration of abstinence from sexual intercourse during the post-partum period? When are women allowed to resume sexual intercourse in your community?
Beliefs about the control and maintenance of the practice of post-partum abstinence	What is your personal experience with post-partum abstinence? Have you been able to maintain abstinence?
Beliefs about factors that facilitate or impede post-partum abstinence	What are some of the ways that the practice is enforced in your community? What are some of the cultural arguments that are used to regulate abstinence? What are some of the myths in that perpetuate sexual abstinence?
Beliefs about positives and negative consequences of post-partum abstinence	What do you think are positive and negative outcomes of this practice?
Demographics	Age, marital status, education, employment, parity, and HIV status

**Table 2 ijerph-16-02590-t002:** Descriptive characteristics of participants.

Variables	Characteristic	Data
Level of education	Primary	2
	Secondary	11
	Tertiary	2
Marital Status	Single	9
	Married	6
Employment status	Employed	3
	Unemployed	12
Place of residence	Peri-urban areas	11
	Rural	3
Age groups	18–30	11
	31–40	4
HIV status	Positive	
	Negative	
Parity	One child	5
	Two children	3
	3–6 children	7
Resumed sexual intercourse	Yes	7
	No	8
Age of baby at resumption	Six months	5
	Three months	1
	Six weeks	1

**Table 3 ijerph-16-02590-t003:** Theory of Planned Behavior (TPB) construct themes

*Theme*	Sub-Theme
Attitude towards post-partum abstinence	
Period of abstinence	
Beliefs about the control of post-partum sexual abstinence	Cultural norms
	Living apart from the partner
	Myths and misconceptions about early resumption
Beliefs about negative consequences of post-partum abstinence	Sexual abstinence is an imposed cultural practice
	Sexual abstinence increases the risk of HIV transmission

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
