# Peer review of "The Role of Culture in Maintaining Post-Partum Sexual Abstinence of Swazi Women"

_ijerph, 2019, doi:10.3390/ijerph16142590_

Round 1

Reviewer 1 Report

This is a really important piece of work. The author should be commended for exploring this issue, both in light of the concerns regarding the transmission of HIV/AIDS, as well as the experiences of women post-partum in this region. 

However, there are a number of concerns with the quality of the work in terms of it being ready for publication. I think with considerable work, this study should be published, but not in its current state.

I have made several comments on the attached document for the author to consider. There are problems with structure, the authors tone/writing style but, the two greatest concerns for me are a) the analysis of the data, this is currently highly descriptive with large quotes used, rather than the researchers interpretation and analysis of the data, and b) I am unsure of the purpose of including women in the sample who are not in a relationship, this contradicts the aim of the paper which is to explore how during abstinence partners seek sex elsewhere and risk transmitting disease.

Author Response

We thank the reviewers for the valuable comments on the manuscript. We have addressed all the comments to best of our ability and understanding of the comments.

Reviewer no 1

It might be more helpful to start with detail about the issue - i.e. unknown risk of transmission of HIV/AIDS during post-partum period - then highlight why this risk is a concern i.e. the practice of sexual abstinence during this period

This opening paragraph is poorly structured; the first three sentences are repetitive and need to be condensed.

This paragraph is also disjointed, several points are being made here; these need separating out.

A little more context to the shift on polygamy is needed, when did this change? Is this across the whole of Africa?

I think this is written in quite a passive manner, the author needs to be clearer about what "attention" means - I am assuming the author is referring to sexual gratification or sexual intimacy, either way, the author needs to write with a little more confidence. Use medical or biological terminology for human functions.

It might help to provide a context regarding the current HIV/AIDS status of Swazi communities

Response: We reviewed the introduction in line with the comments and we used statistics to support and substantiate some of the statements. For example, we provided statistics on the decline of polygamy globally and in the African region and on the HIV statistics in Swaziland. The revised sections are highlighted in blue in the introduction.

The study needs to stand alone - where there other aspects of the dissertation not included in this paper.

Response: This was a mini dissertation of the first author, all the data are presented in the manuscript.

This contradicts the point you made earlier regarding the culture now being monogamous

Response: We show that while polygamy has decreased globally, the culture in Swaziland has not changed substantially in Swaziland.

More of a justification for the choice of analytics is needed - what type of thematic approach was used (TA, Discourse Analysis, Narrative Analysis etc.)

Response: This comment is addressed; we clearly outline the data analysis in line with the other reviewers too.

I am a little confused as to why the women who were single were included here - if your argument is that during a period of abstinence that increases the risk of husbands having sex with other people outside of the marriage, then these women should not form part of the sample and should be excluded.

Response: We clarified that culture prescribes that women, married or single abstain during the post-partum period. Whereas for married women the in laws enforce sexual abstinence, for single women, their mothers exerts the same pressure on them to practice post-partum sexual abstinence.

The detail of the themes needs more work here - perhaps a chart or table demonstrating the links between themes and sub-themes.

Response: We added a table presenting the themes

The data presented in this findings section is highly descriptive and needs greater analysis - this is an example where the raw data is presented with very little analysis - it is not about just presenting a sample of the transcript but an in-depth analysis of what the transcript means.

Response: We reviewed the data presentation in line with the comments from the other reviewers to use a theoretical framework. The revised sections are highlighted in blue in the introduction.

Reviewer 2 Report

The work submitted by Shabangu et al demonstrated that in the Kingdom of Swaizland, “post-partum sexual abstinence is practiced by women in the study setting, despite feeling that the period for post-partum sexual abstinence was long. It’s an interesting article to readers and please find my comment below. 

(1) Did the authors consider marriage as a theme? I find it interesting that 9 out of 15 participants are single. Were they married when they gave birth?  

(2) For section 3.1, it would be appreciated if the authors can make a table listing the status of all 15 participants.  

(3) I find the term “not abstaining” confusing since most of the women have abstained, just not abstaining at the time of interview. Why the authors emphaszie “not abstaining” in the text, for example, Line 164 and 166. 

Author Response

Reviewer no 2

The work submitted by Shabangu et al demonstrated that in the Kingdom of Swaziland, “post-partum sexual abstinence is practiced by women in the study setting, despite feeling that the period for post-partum sexual abstinence was long. It’s an interesting article to readers and please find my comment below.

Did the authors consider marriage as a theme? I find it interesting that 9 out of 15 participants are single. Were they married when they gave birth?

Response: We did not consider marriage as a theme because the cultural prescribes that women, married or single abstain during the post-partum period. We purposely included married and single women to assess the role of culture in perpetuating the practice. The women were married when they gave birth.

It would be appreciated if the authors can make a table listing the status of all 15 participants. 

 Response: We added a table describing the participants

I find the term “not abstaining” confusing since most of the women have abstained, just not abstaining at the time of interview. Why the authors emphasize “not abstaining” in the text, for example, Line 164 and 166.

Response: Thanks for this comment- we have deleted the emphasis on not abstaining in the quotations.

Reviewer 3 Report

This is an interesting article that speaks to culture as a determining factor in understanding sexual abstinence. Results were presented well and so was the discussion section. The paper could be improved with the following suggestions:  

a. Theoretical Framework should be included. Theories worth considering are health behavior theory

b. Recruitment strategies were not included

c. Line 90 lacks a reference

d. What were some examples of the open-ended questions. How were they developed through a study of literature, consultation with community members, theoretically driven. Please explain.

e. Were the transcripts back translated, please discuss.

f. Limitations to the study were not discussed

g. Implications to research, policy and practice were not discussed. This should be within the discussion section. 

Author Response

Reviewer no 3

This interesting article speaks to culture as a determining factor in understanding sexual abstinence. Results were presented well and so was the discussion section. The paper could be improved with the following suggestions:  

Theoretical Framework should be included. Theories worth considering are health behavior theory

Response: We included a theoretical framework and revised the data analysis and presentation to be in line with the constructs of the theoretical framework

Recruitment strategies were not included

Response: We have indicated that the women were selected, using purposive sampling from health facility under the study setting section.

Line 90 lacks a reference

Response: Reference is provided for the population size in Swaziland

What were some examples of the open-ended questions? How were they developed through a study of literature, consultation with community members, theoretically driven? Please explain.

Response: The interview guide was developed after reviewing the literature on post-partum abstinence from studies conducted in the region since there are no studies conducted in Swaziland. The development of the tool was also informed by informal discussions with older women in the community and health professionals. A table with sample questions is added

Were the transcripts back translated, please discuss.

Response: We did not back translate transcripts. Both authors are fluent in the local language and English. The first author translated the transcripts to English and quality and accuracy of the translation was ensured by listening to the audio files and reading the translated transcripts by both authors.

Limitations to the study were not discussed

Response: A section discussing the limitation of the study is included

Implications to research, policy and practice were not discussed. This should be within the discussion section.

Response: We added a section in the conclusion